# GROUP RANK FOR ENCRYPTED DATA

## ABSTRACT

Recently, there has been an increasing demand for privacy-preserving techniques in numerous machine learning algorithms, elevating it to a critical concern. One promising solution involves the application of homomorphic encryption (HE). This study focuses on obtaining statistics based on the ranks of HE-encrypted data as a vital tool for robust data analysis. However, computing ranks in HE comes with significant computational costs due to the necessity of comparison operations, and there is currently no efficient method available. To address this gap, we propose an approximate rank method that exploits pairwise comparisons of data to derive ranks for encrypted information. This method effectively measures the association between two-dimensional ranks. Specifically, by utilizing approximate ranks of two variables, we estimate Spearman rank correlation without relying on perfect sorting and introduce a technique to reduce the number of required comparisons. Numerical experiments have been conducted to validate our approach, demonstrating that the disparity in values between rank correlation and approximate rank correlation is not substantial. Notably, the processing of one block comprising 32,768 ciphertexts took approximately one minute, exhibiting observed linear complexity dependent on the number of blocks.

## 1 INTRODUCTION

The demand for privacy-preserving techniques in numerous machine learning algorithms has been on the rise, emerging as a crucial concern. One potential solution to address this concern is the adoption of homomorphic encryption (HE) (Rivest et al., 1978). HE offers a promising solution for safeguarding data privacy while enabling computations on encrypted data. In our research, we concentrate on acquiring statistics based on the ranks of HE data. However, the computation of ranks in HE involves substantial computational costs due to the requisite comparison operations. Utilizing pairwise comparisons for any two data points allows us to determine the individual rank of encrypted data with a computational complexity of $O(n^2)$, where $n$ is the sample size (Chatterjee & Sengupta, 2015; Cheon et al., 2019; Chatterjee & Sengupta, 2020; Çetin et al., 2021).

To mitigate the complexity associated with finding ranks in HE, the Bitonic sort method (Nakatani et al., 1989) offers a solution with $O\left(n(\log n)^2\right)$ complexity, ensuring a perfect alignment of data. However, this method cannot track the index of encrypted data during sorting, preventing the identification of individual data ranks even after sorting encrypted data. Additionally, the Cheon, Kim, Kim, and Song (CKKS) Scheme (Cheon et al., 2017), the sole HE system capable of handling real numbers, faces challenges in precise comparison operations. Determining the exact rank becomes impossible when given numbers share the same value or exhibit slight differences.

Comparison operations in HE typically involve high computational complexity. This complexity becomes particularly significant when dealing with statistics, such as those in Eq. (2), which are based on ranks of HE data. To address this computational challenge, we propose a method for approximating ranks.

In the case of one-dimensional data, the approximate rank represents ordered groups that categorize and segment the data into grades. Conversely, for two-dimensional data, the approximate rank delineates grades that spatially partition the data. Consider a scenario where the data exist on two dimensions. By combining two marginal approximate ranks, a data point obtains a clearer spatial rank compared to its rank in just one dimension. Leveraging this property, our proposed method effectively evaluates the correlation between ranks in two dimensions. An advantageous feature of

this method is that, as the dimensionality increases, it enables more accurate identification of the location of data points in multi-dimensional space. This capability proves effective in measuring the correlation between ranks, offering a valuable tool for analyzing complex data structures.

## 1.1 CONTRIBUTIONS

This paper presents a method for efficient approximate computation based on the group rank of homomorphic encryption (HE). The main idea is to introduce a grouped rank within HE, allowing us to estimate ranks that are HE-friendly for large-sized HE data.

In order to improve the computational complexity, currently at $O(n^2)$, required for precise ranking in homomorphic encryption, we introduce the notion of grouped ranks, denoted as 'group rank.' This concept proves effective in estimating correlations among variables in multidimensional space, even though the approximation accuracy is substantial in one-dimensional space. To the best of our knowledge, this work represents the first exploration in homomorphic encryption. The proposed method specifically tackles the challenge of estimating Spearman rank correlation (Kruskal, 1958) in two-dimensional space, showcasing a reduction in estimation error. Additionally, the utilization of group rank in homomorphic encryption suggests the potential for facilitating HE-friendly computations based on ranks for various statistical estimation problems. Furthermore, we enhance speed by implementing an efficient memory utilization of the comparison operations.

The rest of the paper is organized as follows. Section 2 reviews HE shortly. Section 3 describes the proposed method for estimating ranks. Then, Section 4 presents a simulation study and real data analysis. Section 5 concludes and discusses future works.

## 1.2 RELATED WORKS

HE enables computations on encrypted data without decryption, facilitating data analysis while preserving data privacy. The concept of HE was initially proposed by Rivest et al. (1978), and Gentry (2009) demonstrated the existence of homomorphic encryption schemes that allow for unlimited multiplications. Subsequently, several HE schemes have been developed. Cheon et al. (2017) proposed an approximate HE scheme that supports particular fixed-point arithmetic commonly referred to as block floating-point arithmetic, known as the Cheon, Kim, Kim, and Song (CKKS) scheme. The CKKS scheme is recognized as one of the most efficient HE schemes that support computation on real/complex data. Unlike other HE schemes designed for integer (Brakerski et al., 2012; Fan & Vercauteren, 2012) or binary (Chillotti et al., 2016) messages, the CKKS scheme is intended for real/complex messages.

## 2 PRELIMINARIES

This section introduces homomorphic encryption (HE), the CKSS scheme, and Spearman rank correlation briefly.

### HOMOMORPHIC ENCRYPTION

HE is a class of encryption schemes that enables computation over encrypted data. Fully HE refers to an encryption system that preserves operations such as addition and multiplication in an encrypted state. That is, for encryption homomorphism (Enc) and decryption homomorphism (Dec), the following holds.

$$\text{Dec}(\text{Enc}(x) + \text{Enc}(y)) = x + y,$$
$$\text{Dec}(\text{Enc}(x) \cdot \text{Enc}(y)) = x \cdot y.$$

To estimate the ranking, comparison operations between data are necessary. For example, the 0-1 function with a single discontinuity point is commonly used to compare two numbers for finding various statistics. However, since the CKSS scheme is designed to handle only polynomial functions, one must approximate non-polynomial functions with polynomial ones. To approximate the 0-1 function, the CKKS scheme employs a rational function that can be represented as the Taylor series, leading to time-consuming calculations in HE (Cheon et al., 2019; 2020). Their algorithms achieve

optimality regarding asymptotic computational complexity among polynomial approximations for min/max and comparison operations.

SPEARMAN RANK CORRELATION

The coefficient of rank correlation is a measure of association between ranks. Let $X$ and $Y$ be random variables of some probability distributions. A random sample of $n$ pairs

$$(X_1, Y_1), (X_2, Y_2), \ldots, (X_n, Y_n)$$

is drawn from a bivariate population. Given the sample, the Pearson correlation coefficient (Gibbons & Chakraborti, 2020) is

$$\hat{\rho}(X, Y) = \frac{\sum_i \left(X_i - \bar{X}\right)\left(Y_i - \bar{Y}\right)}{\sqrt{\sum_i \left(X_i - \bar{X}\right)^2 \sum_i \left(Y_i - \bar{Y}\right)^2}}.$$

As this coefficient relies on the second moment, it is generally not considered a robust statistic. An alternative approach to constructing a robust statistic involves considering rank information rather than the actual values.

A rank $r : X \in \mathbb{R} \longrightarrow \{1, 2, \ldots, n\}$ is a function from $X$ to an integer value. The indicator function $I$ is defined as follows: for a condition $C$, $I(C)$ is 1 if $C$ is true, and 0 if it is not. Then, in a sample, the rank is given as

$$r(X) = \sum_{j=1}^{n} I(X_j \leq X). \tag{1}$$

In each sample, let $r_i = r(X_i)$ and $s_i = r(Y_i)$ be the ranks of $X_i$ and $Y_i$, respectively. Given the observed paired data $\{(X_i, Y_i) | i = 1, 2, \ldots, n\}$, the Spearman rank correlation coefficient is defined as

$$\hat{\rho}(r(X), r(Y)) = \frac{\sum_{i=1}^{n}(r_i - \bar{r})(s_i - \bar{s})}{\sqrt{\sum_{i=1}^{n}(r_i - \bar{r})^2}\sqrt{\sum_{i=1}^{n}(s_i - \bar{s})^2}} \tag{2}$$

where $r_i$ and $s_i$ are the ranks of $\{X_1, \ldots, X_n\}$ and $\{Y_1, \ldots, Y_n\}$, respectively. Also, denote $\bar{r}$ and $\bar{s}$ are the sample means of the ranks. Note that if there are no ties, either $\bar{r}$ or $\bar{s}$ is simply $(n+1)/2$ and $\sum_{i=1}^{n} r_i = \sum_{i=1}^{n} s_i = \sum_{i=1}^{n} i = n(n+1)/2$. Furthermore, in the denominator

$$\sum_{i=1}^{n}(r_i - \bar{r})^2 = \sum_{i=1}^{n}(s_i - \bar{s})^2 = \sum_{i=1}^{n}\left(i - \frac{n+1}{2}\right)^2,$$

which leads to a constant $\frac{n(n-1)(n+1)}{12}$.

The Spearman rank correlation coefficient, denoted as $R$, between two variables is calculated as the Pearson correlation using the rank values of those variables. This is considered a distribution-free statistic. In contrast, Pearson's correlation assesses linear relationships. When there are no ties in data values, a Spearman rank correlation of $\pm 1$ implies a perfect monotone relationship between the two variables.

DATA PACKING

A one-dimensional vector, including many independent data, can be encrypted in a single ciphertext, which is referred to as packing to be performed in parallel. Each block consists of 32,768 slots, and a block comparison with univariate values is approximately equivalent to one slot, resulting in high computational efficiency.

## 3 PROPOSED METHOD

Let $F$ and $G$ be cumulative distribution functions (CDF). The expectation of the Spearman rank correlation coefficient $R$, which is called grade correlation coefficient (Gibbons & Chakraborti, 2020), is given as

$$\lim_{n \to \infty} \mathrm{E}(R) = \rho(F(X), G(Y)),$$

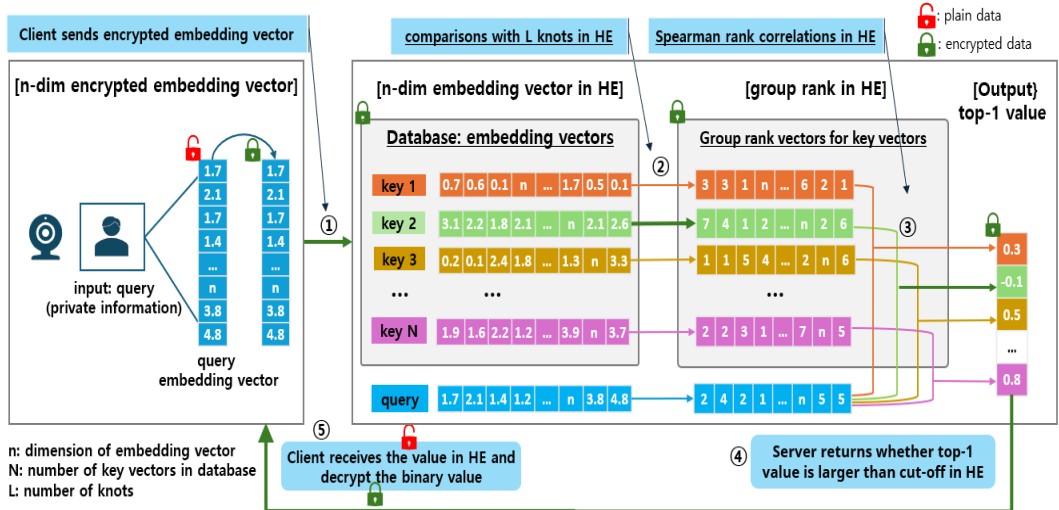

Figure 1: An illustrative practical scenario about face recognition using Spearman rank correlation

and $\hat{\rho}(r(X), r(Y))$ in Eq. (2) is its unbiased estimator in large samples. This estimation requires two marginal CDFs of having complexity $O(n^2)$ comparisons.

Thus, to reduce computational complexity, we propose a method to estimate the Spearman rank correlation coefficient with lower computational cost, primarily by focusing on reducing the number of comparison operations. The key idea is to utilize the CDF values to determine the ranks of the data points. Instead of performing comparisons on the entire dataset, the comparisons are limited to the CDF values at specifically designed points that are uniformly distributed over intervals with a length of $L$.

In the domain of HE applied to machine learning and artificial intelligence, previous studies have demonstrated its efficacy in various contexts such as deploying logistic regression models for inference (Kim et al., 2018) or encrypting models like ResNet-24 for privacy-preserving inference (Lee et al., 2022). Moreover, in the context of Machine Learning as a Service(MLaaS), techniques such as transfer learning involve encrypting client data to update models on cloud servers securely (Lee et al., 2023). Figure 1 illustrates the use of Spearman rank correlations in the face recognition problem: (1) In a communication system between the client and server, the encrypted embedding vector of a face image is sent to the server. (2) Through comparisons with $L$ knots, the embedding vector is transformed into a group rank in HE. (3) The server calculates the Spearman rank correlations in HE between the group rank vectors of the query and key vectors. (4) The server then returns whether the top-1 value is larger than a predefined cut-off in HE. (5) The client receives the result in HE and decrypts the binary value. This process ensures that sensitive information remains encrypted, addressing privacy concerns.

### 3.1 GROUP RANK

By confining the operations to the CDF values of the designed points, we aim to reduce the computational cost associated with estimating the Spearman rank correlation. Moreover, the proposed method is designed for parallel execution, thereby enhancing efficiency. If there are $s$-SIMD (single instruction and multiple data) blocks, the total number of comparisons would be $sL$. The results of these comparisons are stored in a matrix with dimensions $(32, 768 \times s) \times L$.

Let $\Pr(E)$ represent the probability of an event $E$. For a positive integer $L$, we define a knot of length $L$ by a finite sequence $\xi = \{\xi_j\}_{j=1}^{L}$ of real numbers such that $\xi_1 < \xi_2 < \cdots < \xi_L$. This definition is detailed further as follows: Let $\xi$ and $\eta$ be knots of $X$. We say that $\eta$ is a finer than $\xi$ if, for any $i$, there exists $j$ such that $(\eta_j, \eta_{j+1})$ is a subset of $(\xi_i, \xi_{i+1})$, and there exist $i_0, j_0$, and

$(\eta_{j_0}, \eta_{j_0+1})$ that are strictly subsets of $(\xi_{i_0}, \xi_{i_0+1})$. As $L$ increases, a sequence of knots exists that converges to $\xi$, such that every interval of $\xi$ contains at most one element of $X$.

Let $\{X_{(1)}, \ldots, X_{(n)}\}$ be order statistics of $\{X_1, \ldots, X_n\}$ such that $X_{(1)} \leq X_{(2)} \leq \cdots \leq X_{(n)}$. Let the empirical CDF (ECDF) at $x$ be

$$F_n(x) = \frac{1}{n} \sum_{i=1}^{n} I(X_i \leq x),$$

and a lower bound of $X$ be $\xi_{\min} = -\infty$. We assume that knots are within the range of the observed data, such as $X_{(1)} < \xi_1$ and $\xi_L < X_{(n)}$.

**Definition 3.1.** We define an approximate ECDF of $X$ for knot $\xi$ of length $L$ as

$$F(x; \xi) = \sum_{j=1}^{L} I(\xi_j \leq x) \hat{\Pr}(\xi_{j-1} \leq X < \xi_j), \tag{3}$$

where $\xi_0 = \xi_{min}$ and $\hat{\Pr}(\xi_{j-1} \leq X < \xi_j) = \frac{1}{n} \sum_{i=1}^{n} I(\xi_{j-1} \leq X_i < \xi_j)$.

Proposition 3.2 demonstrates that $F(x; \xi)$ can be an approximation of ECDF.

**Proposition 3.2.** *Let $\{X_1, \cdots, X_n\}$ be a given data and a knot $\xi$ of length $L$. Then, for any $x \in \mathbb{R}$,*

$$F(x; \xi) = F_n(\xi_* -),$$

*where $\xi_*(x) = \max_{1 \leq j \leq L}\{\xi_j \,|\, \xi_j \leq x\}$. Furthermore, if $\eta$ is a knot of $\{X_1, \cdots, X_n\}$ satisfying $\xi \subset \eta$ then*

$$F(x; \xi) \leq F(x; \eta) \leq F_n(x). \tag{4}$$

Comparing only with $L$ knot points, we obtain $F(X_1; \xi), \ldots, F(X_n; \xi)$. As by-products, we define an approximate rank of $X$, for $i = 1, \ldots, n$,

$$r(X_i; \xi) = 1 + nF(X_i; \xi).$$

This can be regarded as an estimate of rank. When $L < n$, there are $L$ distinct group ranks. If there is no confusion, we denote $F(x) := F(x; \xi)$ and $r(x) := r(x; \xi)$.

Note that the definition in Eq. equation 3 implies that $F(x; \xi) < 1$ for $\xi_L < x$. In other words,

$$\lim_{x \to \infty} F(x; \xi) < 1.$$

Supposing all the data are distinct, let us assume that $\xi_L$ falls within the interval $X_{(n-1)} < \xi_L < X_{(n)}$. According to Eq. equation 3, this implies that $F(X_{(n)}; \xi) = 1 - 1/n$. In this case, $F(X_{(n)} + \epsilon; \xi) = 1 - 1/n$ for $\epsilon > 0$. Therefore, to become an adequate estimator of the CDF, it requires adding a $\xi_{\max}$ which is greater than $X_{(n)}$ to the knot $\xi$. This leads to the definition in Eq. equation 3, which is defined on an expanded knot $\xi' = \xi \cup \xi_{\max}$. However, when finding group rankings, there is no need to compare with $\xi_{\max}$, so we have restricted the range of $\xi$ to the range of the data.

For example, assume that there are two knots, $\xi_1$ and $\xi_2$, where $\xi_{\min}$ and $\xi_{\max}$ represent the lower and upper bounds of data, respectively. If $x$ lies in $[\xi_2, \xi_{\max})$, then the result of the comparison $x \geq \xi_2$ is 1. Similarly, the results of the comparison $x \geq \xi_1$ is also 1. Consequently, the result vector is $(1, 1)$ for this observation. Furthermore, the comparison vectors with $[\xi_1, \xi_2)$ or $(\xi_{\min}, \xi_1)$ are $(1, 0)$ or $(0, 0)$, respectively. If $x \in [\xi_1, \xi_2)$, then the indicators of the range are $(1, 0)$. If their cumulative functions are $(0.7, 0.2, 0.1)$, then its empirical CDF value is $1 \cdot 0.7 + 0 \cdot 0.2 = 0.7$. Table 1 shows the comparison results when data are sorted.

The boolean output of comparisons with $\{\xi_i\}_{i=1}^{L}$ are encrypted, rendering each comparison with $\xi_k$ unknown. However, the sum of these, $\sum_{i=1}^{n} I(X_i > \xi_k)$, forms a sufficient statistic for finding the rank. To efficiently utilize memory in storing the comparison results, we opt to store them sequentially in a list rather than using a matrix storage method. The algorithm for calculating group ranks is presented in Algorithm 1.

If a finer set lacks a hierarchical structure, the definition in Eq. equation 3 does not ensure inequality. This implies that empirical CDF is not monotonic in knots. Hence, we assume the case with a hierarchical structure. The following proposition describes the monotonic property of group rank according to a finer knot set.

Table 1: Result of comparisons with knots

| $X$ | $\xi_1$ | $\cdots$ | $\xi_l$ | $\cdots$ | $\xi_L$ | RANK |
|---|---|---|---|---|---|---|
| $\in (\xi_{\min}, \xi_1)$ | 0 | $\cdots$ | 0 | $\cdots$ | 0 | 1 |
| $\vdots$ | | | | | | |
| $\in [\xi_l, \xi_{l+1})$ | 1 | $\cdots$ | 1 | $\cdots$ | 0 | $n - a_l + 1$ |
| $\vdots$ | | | | | | |
| $\in [\xi_L, \xi_{\max})$ | 1 | $\cdots$ | 1 | $\cdots$ | 1 | $n - a_L + 1$ |
| SUM | $a_1$ | $\cdots$ | $a_l$ | $\cdots$ | $a_L$ | |

---

**Algorithm 1** Algorithm for finding group rank, $\mathsf{grank}(\mathbf{x}, \xi)$

---

1: Input: ciphertext $\mathbf{x}$ with $L$ knots, $\xi$
2: Output: group rank of $\mathbf{x}$.
3: **Initialization** $\mathsf{grank}(x_i, \xi) = 1$,
4: $a_0 = n$ $\hspace{6cm}$ $n \cdot \Pr(X \geq -\infty)$
5: **for** $k = 1 : L$ **do**
6: $\quad$ **for** $i = 1, \cdots, n$ **do**
7: $\quad\quad$ $\mathrm{comp}_{i,k} = 1$ if $x_i \geq \xi_k$ else 0, $\hspace{3cm}$ $I(x_i \geq \xi_k)$
8: $\quad\quad$ $a_k = \sum_i \mathrm{comp}_{i,k}$ $\hspace{4cm}$ $n \cdot \Pr(X \geq \xi_k)$
9: $\quad\quad$ $b_k = a_k - a_{k-1}$ $\hspace{3cm}$ $n \cdot \Pr(\xi_{k-1} \leq X < \xi_k)$
10: $\quad\quad$ $\mathsf{grank}(x_i, \xi) = \mathsf{grank}(x_i, \xi) + b_k \cdot \mathrm{comp}_{i,k}$
11: $\quad$ **end for**
12: **end for**
13: **Return** $\mathsf{grank}$

---

**Proposition 3.3.** *Let $X$ be given data. Suppose $\xi$ and $\eta$ are knots of $X$ such that $\xi \subset \eta$. Let $\{r(X_i; \xi)\}$ and $\{r(X_i; \eta)\}$ be the group rank of $X$ with respect to $\xi$ and $\eta$, respectively. Then,*

$$r(X_i; \xi) \leq r(X_i; \eta) \leq r(X_i).$$

*Furthermore, if $X$ has no ties, then $\bar{r}_\xi \leq \bar{r}_\eta \leq (n+1)/2$ where $\bar{r}_\xi$ and $\bar{r}_\eta$ are the means under knot sets $\xi$ and $\eta$, respectively.*

If we assume that the data are distinct, Proposition 3.3 implies that the upper bound approaches $(n+1)/2$ as the knots form a finer set. Since the ECDF exhibits jumps of size $1/n$ at distinct data points, the following theorem can be readily proved.

**Theorem 3.4.** *With the knot $\xi_i = X_i, i = 1, \ldots, n,$, the approximate rank $r(X_i; \xi) = 1 + nF(X_i; \xi)$ is equivalent to the rank $r(X)$ in Eq. equation 1. When the number of elements in the group is 1, the group rank becomes the usual rank.*

Theorem 3.4 implies that when using $n$ knots, $O(n^2)$ comparisons are needed.

### 3.2 RANK CORRELATION ON TWO GROUP RANKS

When comparing knots and data, we obtain the group ranks. Similarly, if $G(\cdot; \eta)$ is the approximate ECDF of $Y$ for a knot $\eta = \{\eta_j\}_{j=1}^L$ of length $L$, we obtain $G(Y_1; \eta), \ldots, G(Y_n; \eta)$, resulting in approximate ranks $r(Y_i; \eta) = 1 + nG(Y_i; \eta)$, $i = 1, \ldots, n$. From $\{(X_i, Y_i) | i = 1, \ldots, n\}$, we obtain pairs of ranks

$$\{(r(X_i; \xi), r(Y_i; \eta)) | i = 1, \ldots, n\},$$

without the need for sorting operations, and subsequently an estimate $\hat{\rho}(r(X), r(Y))$. By obtaining paired integer ranks from group ranks in the data consisting of $n$ sets, we can define an estimator of the Spearman rank correlation coefficient in two dimensions. Algorithm 2 illustrates that Spearman rank correlation coefficient is based on two group ranks.

As the dimension increases, distances between data points also increase, resulting in sparsity. This implies that for a given point $\mathbf{x}$ and precision $\epsilon$, the probability of observing other points in the neighborhood $\{\mathbf{x}' : |\mathbf{x} - \mathbf{x}'| < \epsilon\}$ decreases. For example, for a number $x \in [0, 1]$, the distance $d(x, z)$ is

---

**Algorithm 2** Spearman rank correlation based on two group ranks

---
1: Input: two ciphertexts $\mathbf{x}$ and $\mathbf{y}$ having $\xi^X$ and $\xi^Y$ with $L$ knots respectively
2: Output: Spearman rank correlation of two ciphertexts $\mathbf{x}$ and $\mathbf{y}$.
3: $r(\mathbf{x}; \xi^X) \longleftarrow \mathsf{grank}(\mathbf{x}, \xi^X)$
4: $r(\mathbf{y}; \xi^Y) \longleftarrow \mathsf{grank}(\mathbf{y}, \xi^Y)$.
5: Calculate Spearman rank correlation $\hat{\rho}$ with $r(\mathbf{x}; \xi^X)$ and $r(\mathbf{y}; \xi^Y)$.
6: **Return** $\hat{\rho}$

---

approximately 1/3 for any $x' \in [0, 1]$. If $\mathbf{x}, \mathbf{x}' \in [0, 1]^2$, the distance $d(\mathbf{x}, \mathbf{x}')$ is about 0.521 for any $\mathbf{x}' \in [0, 1]^2$. Thus, as the dimension grows, the average distance between arbitrary points within a single unit cell increases by approximately 1.6 times compared to one dimension. Despite potential errors in rank estimates for each dimension, the probability of observing other points around a point in two dimensions is lower, making the spatial location clearer in high dimensions. This is evident in the following uniform distribution.

Let $X_i$ be random variables in a $d$-dimensional uniform distribution and the range for each dimension is $[0, 1]$. Denote a random sample $S_d = \{X_i = (X_{i1}, \cdots, X_{id})^T \in \mathbb{R}^d\}_{i=1}^n$ of $n$ with $d$ dimension. We define the minimum distance of $S_d$ as

$$\min S_d := \min_{X_i, X_j \in S_d} \mathrm{dist}(X_i, X_j)$$

where $\mathrm{dist}(X_i, X_j)$ is the distance between $X_i$ and $X_j$ in $S_d$. Then, for any $d > 0$,

$$\min S_{d+1} \geq \min S_d.$$

In two dimensions, there are $L^2$ blocks where each cell indicates a pair of two ranks spatially. Furthermore, when $L = n$, the spatial rank is unique. This implies that two spatial ranks for any two cells have at least one different marginal rank, either $r(X_i; \xi^X)$ or $r(Y_i; \xi^Y)$. We claim that the group rank in two dimensions becomes clearer than in one dimension.

To investigate the characteristics of the group rank in two dimensions based on the definition of one dimension, as discussed in Section 3.1, we consider $\eta$ as a finer knot set of $\xi$. Denote $\xi^X, \eta^X$ and $\xi^Y, \eta^Y$ as the knot sets for variables $X$ and $Y$, respectively. This implies $L = |\xi| < |\eta| = L'$. Let $r(X_i; \xi^X)$ and $r(Y_i; \eta^Y)$ be marginal ranks for each variable given knot set $\xi$ and its finer knot set $\eta$, respectively. Taking two indices $i, j (i \neq j)$, denote pairs of ranks in two dimension as follows: $r(i; \xi^X, \xi^Y) = (r(X_i; \xi^X), r(Y_i; \xi^Y))$ and $r(j; \xi^X, \xi^Y) = (r(X_j; \xi^X), r(Y_j; \xi^Y))$. Then, the following inequality holds:

$$\begin{aligned}
&\|r(i; \xi^X, \xi^Y) - r(j; \xi^X, \xi^Y))\|_1 \\
\leq\ &\|r(i; \xi^X, \eta^Y) - r(j; \xi^X, \eta^Y))\|_1 \\
&\text{or } \|r(i; \eta^X, \xi^Y) - r(j; \eta^X, \xi^Y))\|_1 \\
\leq\ &\|r(i; \eta^X, \eta^Y), r(j; \eta^X, \eta^Y))\|_1,
\end{aligned}$$

where $\|\cdot\|_1$ is the $l_1$ norm.

If either knots become finer or $L$ increases, the distance between two distinct data points increases. Although they may have the same rank in one dimension, there is a higher likelihood of having different ranks in two dimensions. With increased dimensionality, we can anticipate a clearer spatial ranking and expect fewer errors in estimates based on combined ranks. In conclusion, spatially different rankings between two samples can be more clearly discerned in multivariate cases than univariate ones.

## 4 NUMERICAL STUDY

This section investigates the performance of the Spearman rank correlation coefficient estimation method using the proposed group rank through simulated and real data.

We utilized the HEaaN library, which operates under the CKKS scheme and is accessible from https://heaan.it/. This library comprises HEaaN (C++ library) and HEaaN.STAT (python

library). These parameters include $\log(\text{ring dimension}) = 16$, hamming weight=192, and standard deviation of the Gaussian distribution=3.2. The number of slots is $32,768 = \log(\text{ring dimension}/2)$. Additionally, after bootstrapping, the ciphertext level reaches 12, with the minimum level for bootstrapping set at 3, enabling 9 multiplications between each bootstrap operation. For further information, refer to Cheon et al. (2017).

## 4.1 SIMULATION

We generate $n$ pairs of samples of $x$ and $y$ by considering distributions for each random variable, including $N(0, 1)$, log-normal, and chi-squared($\chi^2$) distributions. Then, we create four combinations (normal, normal), (normal, log-normal), (normal, $\chi^2$), and (log- normal, $\chi^2$).

Table 2: In (normal, normal) and (normal, log-normal) settings, MAD for Homomorphic encryption with $n = 32768 \times s$. The standard deviation is in parenthesis.

| $s$ | $L$ | (NORMAL, NORMAL) | | (NORMAL, LOG-NORMAL) | |
|---|---|---|---|---|---|
| | | UNIFORM | RANDOM | UNIFORM | RANDOM |
| 1 | 16 | $0.0099_{(0.0014)}$ | $0.0141_{(0.0057)}$ | $0.0041_{(0.0028)}$ | $0.0010_{(0.0007)}$ |
| | 32 | $0.0028_{(0.0008)}$ | $0.0040_{(0.0016)}$ | $0.0034_{(0.0022)}$ | $0.0005_{(0.0003)}$ |
| | 64 | $0.0007_{(0.0003)}$ | $0.0014_{(0.0008)}$ | $0.0018_{(0.0013)}$ | $0.0003_{(0.0004)}$ |
| | 128 | $0.0002_{(0.0001)}$ | $0.0004_{(0.0003)}$ | $0.0007_{(0.0005)}$ | $0.0002_{(0.0002)}$ |
| 2 | 16 | $0.0116_{(0.0018)}$ | $0.0153_{(0.0042)}$ | $0.0034_{(0.0025)}$ | $0.0006_{(0.0004)}$ |
| | 32 | $0.0028_{(0.0012)}$ | $0.0053_{(0.0024)}$ | $0.0017_{(0.0015)}$ | $0.0002_{(0.0003)}$ |
| | 64 | $0.0010_{(0.0009)}$ | $0.0014_{(0.0010)}$ | $0.0016_{(0.0011)}$ | $0.0002_{(0.0002)}$ |
| | 128 | $0.0008_{(0.0005)}$ | $0.0009_{(0.0006)}$ | $0.0006_{(0.0006)}$ | $0.0001_{(0.0001)}$ |
| 3 | 16 | $0.0118_{(0.0017)}$ | $0.0133_{(0.0040)}$ | $0.0026_{(0.0015)}$ | $0.0005_{(0.0005)}$ |
| | 32 | $0.0026_{(0.0015)}$ | $0.0046_{(0.0029)}$ | $0.0020_{(0.0015)}$ | $0.0004_{(0.0003)}$ |
| | 64 | $0.0014_{(0.0011)}$ | $0.0017_{(0.0013)}$ | $0.0012_{(0.0010)}$ | $0.0002_{(0.0001)}$ |
| | 128 | $0.0010_{(0.0007)}$ | $0.0011_{(0.0007)}$ | $0.0008_{(0.0007)}$ | $0.0001_{(0.0001)}$ |
| 4 | 16 | $0.0115_{(0.0018)}$ | $0.0139_{(0.0046)}$ | $0.0023_{(0.0015)}$ | $0.0005_{(0.0003)}$ |
| | 32 | $0.0023_{(0.0013)}$ | $0.0039_{(0.0021)}$ | $0.0019_{(0.0013)}$ | $0.0003_{(0.0002)}$ |
| | 64 | $0.0013_{(0.0012)}$ | $0.0019_{(0.0017)}$ | $0.0008_{(0.0007)}$ | $0.0002_{(0.0001)}$ |
| | 128 | $0.0013_{(0.0009)}$ | $0.0014_{(0.0011)}$ | $0.0004_{(0.0004)}$ | $0.0001_{(0.0001)}$ |

Table 3: In (normal, $\chi^2$) and (log- normal, $\chi^2$) settings, MAD for Homomorphic encryption with $n = 32768 \times s$. The standard deviation is in parenthesis.

| $s$ | $L$ | (NORMAL, $\chi^2$) | | (LOG- NORMAL, $\chi^2$) | |
|---|---|---|---|---|---|
| | | UNIFORM | RANDOM | UNIFORM | RANDOM |
| 1 | 16 | $0.0027_{(0.0017)}$ | $0.0010_{(0.0005)}$ | $0.0054_{(0.0039)}$ | $0.0011_{(0.0008)}$ |
| | 32 | $0.0017_{(0.0014)}$ | $0.0006_{(0.0005)}$ | $0.0034_{(0.0027)}$ | $0.0006_{(0.0006)}$ |
| | 64 | $0.0014_{(0.0009)}$ | $0.0003_{(0.0003)}$ | $0.0017_{(0.0011)}$ | $0.0004_{(0.0003)}$ |
| | 128 | $0.0009_{(0.0007)}$ | $0.0002_{(0.0001)}$ | $0.0015_{(0.0012)}$ | $0.0002_{(0.0001)}$ |
| 2 | 16 | $0.0023_{(0.0017)}$ | $0.0007_{(0.0005)}$ | $0.0038_{(0.0026)}$ | $0.0005_{(0.0004)}$ |
| | 32 | $0.0013_{(0.0010)}$ | $0.0005_{(0.0003)}$ | $0.0022_{(0.0016)}$ | $0.0003_{(0.0002)}$ |
| | 64 | $0.0009_{(0.0007)}$ | $0.0002_{(0.0002)}$ | $0.0015_{(0.0013)}$ | $0.0003_{(0.0002)}$ |
| | 128 | $0.0006_{(0.0005)}$ | $0.0001_{(0.0001)}$ | $0.0011_{(0.0013)}$ | $0.0001_{(0.0001)}$ |
| 3 | 16 | $0.0017_{(0.0012)}$ | $0.0006_{(0.0004)}$ | $0.0031_{(0.0026)}$ | $0.0004_{(0.0003)}$ |
| | 32 | $0.0013_{(0.0010)}$ | $0.0003_{(0.0002)}$ | $0.0025_{(0.0015)}$ | $0.0003_{(0.0002)}$ |
| | 64 | $0.0008_{(0.0008)}$ | $0.0002_{(0.0001)}$ | $0.0015_{(0.0011)}$ | $0.0001_{(0.0001)}$ |
| | 128 | $0.0005_{(0.0003)}$ | $0.0001_{(0.0001)}$ | $0.0009_{(0.0006)}$ | $0.0001_{(0.0001)}$ |
| 4 | 16 | $0.0019_{(0.0014)}$ | $0.0005_{(0.0006)}$ | $0.0032_{(0.0024)}$ | $0.0006_{(0.0005)}$ |
| | 32 | $0.0013_{(0.0011)}$ | $0.0003_{(0.0003)}$ | $0.0023_{(0.0018)}$ | $0.0003_{(0.0003)}$ |
| | 64 | $0.0007_{(0.0006)}$ | $0.0002_{(0.0001)}$ | $0.0014_{(0.0010)}$ | $0.0001_{(0.0001)}$ |
| | 128 | $0.0004_{(0.0003)}$ | $0.0001_{(0.0001)}$ | $0.0008_{(0.0005)}$ | $0.0001_{(0.0001)}$ |

We report the differences between the actual Spearman rank correlation coefficient and the estimate by repeating the process 20 times. We define $L$ knots as 16, 32, 64, and 128. Additionally, we explore two knot selection methods: the uniform selection method and random selection. As an evaluation metric, we compute the mean absolute difference and standard deviation between actual and estimated values for each experimental dataset.

Table 2 and 3 displays the MAD (mean absolute difference). Overall, it can be seen that as the number of knots increases, the MAD value decreases. In particular, with 64 and 128 knots, it is evident that they exhibit almost identical performance.

In the case of (normal, normal), the uniform selection method shows superior performance compared to the random selection method. However, when at least one of the two variables is non-normal, the random selection method performs better. Notably, the random selection with $L = 16$ knots performs similarly or even better than the uniform selection with $L = 128$. Unlike the considered distributions, the normal distribution is symmetric, and thus, the uniform knot selection method evenly allocates the number of samples within each group rank. This characteristic allows it to produce smaller MAD values when the number of knots is small, compared to the random selection method. However, as the value of $L$ increases, the difference diminishes, and with $L = 128$, both methods exhibit similar performance.

On the other hand, in the case of (log-normal, $\chi^2$), where both variables do not follow a normal distribution, the performance gap between uniform and random selections becomes more pronounced. Since it is challenging to specify the distribution in HE data beforehand, the choice of knot selection can significantly impact the accuracy of the estimation. Therefore, the method requires a data-driven optimal knot selection. It is suggested that the random selection of knots is likely the most suitable method for this purpose.

Table 4: Elapsed time for homomorphic encryption with $n = 32768 \times s$. The standard deviation is in parenthesis.

| | | (NORMAL, NORMAL) | | (NORMAL, LOG-NORMAL) | | (NORMAL, $\chi^2$) | | (LOG- NORMAL, $\chi^2$) | |
|---|---|---|---|---|---|---|---|---|---|
| $s$ | $L$ | UNIFORM | RANDOM | UNIFORM | RANDOM | UNIFORM | RANDOM | UNIFORM | RANDOM |
| 1 | 16 | $9.68_{(0.06)}$ | $15.22_{(0.09)}$ | $9.72_{(0.06)}$ | $15.34_{(0.05)}$ | $9.69_{(0.06)}$ | $15.25_{(0.08)}$ | $9.58_{(0.020)}$ | $15.09_{(0.02)}$ |
| | 32 | $17.50_{(0.13)}$ | $25.96_{(0.15)}$ | $17.82_{(0.28)}$ | $26.42_{(0.44)}$ | $17.36_{(0.10)}$ | $25.82_{(0.17)}$ | $17.30_{(0.00)}$ | $25.67_{(0.06)}$ |
| | 64 | $33.17_{(0.35)}$ | $45.54_{(0.58)}$ | $33.13_{(0.19)}$ | $45.43_{(0.16)}$ | $32.87_{(0.13)}$ | $45.00_{(0.18)}$ | $32.64_{(0.06)}$ | $44.74_{(0.05)}$ |
| | 128 | $64.26_{(0.59)}$ | $81.22_{(0.57)}$ | $63.92_{(0.58)}$ | $80.84_{(0.64)}$ | $63.90_{(0.27)}$ | $80.79_{(0.19)}$ | $63.38_{(0.06)}$ | $80.32_{(0.14)}$ |
| 2 | 16 | $17.74_{(0.07)}$ | $23.25_{(0.08)}$ | $17.58_{(0.04)}$ | $23.09_{(0.03)}$ | $17.69_{(0.12)}$ | $23.19_{(0.08)}$ | $17.52_{(0.04)}$ | $23.02_{(0.04)}$ |
| | 32 | $33.12_{(0.10)}$ | $41.54_{(0.09)}$ | $32.91_{(0.23)}$ | $41.30_{(0.29)}$ | $32.98_{(0.13)}$ | $41.44_{(0.16)}$ | $32.74_{(0.06)}$ | $41.08_{(0.07)}$ |
| | 64 | $63.80_{(0.16)}$ | $75.92_{(0.11)}$ | $63.45_{(0.31)}$ | $75.39_{(0.17)}$ | $63.69_{(0.30)}$ | $75.66_{(0.22)}$ | $63.09_{(0.08)}$ | $75.11_{(0.07)}$ |
| | 128 | $125.05_{(0.22)}$ | $141.95_{(0.22)}$ | $124.45_{(0.51)}$ | $141.25_{(0.44)}$ | $124.95_{(0.51)}$ | $141.70_{(0.47)}$ | $124.00_{(0.00)}$ | $140.75_{(0.44)}$ |
| 3 | 16 | $25.83_{(0.08)}$ | $31.32_{(0.09)}$ | $25.55_{(0.08)}$ | $31.14_{(0.14)}$ | $25.61_{(0.11)}$ | $31.18_{(0.18)}$ | $25.50_{(0.02)}$ | $30.96_{(0.05)}$ |
| | 32 | $48.84_{(0.12)}$ | $57.09_{(0.12)}$ | $48.42_{(0.21)}$ | $56.79_{(0.21)}$ | $48.50_{(0.26)}$ | $56.97_{(0.26)}$ | $49.81_{(6.63)}$ | $56.71_{(0.12)}$ |
| | 64 | $94.72_{(0.22)}$ | $106.75_{(0.44)}$ | $94.00_{(0.29)}$ | $106.05_{(0.22)}$ | $94.35_{(0.32)}$ | $106.10_{(0.31)}$ | $93.92_{(0.09)}$ | $106.00_{(0.00)}$ |
| | 128 | $186.30_{(0.47)}$ | $202.80_{(0.41)}$ | $185.35_{(0.49)}$ | $202.15_{(0.81)}$ | $187.35_{(1.63)}$ | $204.50_{(1.99)}$ | $185.00_{(0.00)}$ | $201.40_{(0.50)}$ |
| 4 | 16 | $33.92_{(0.12)}$ | $39.40_{(0.10)}$ | $33.60_{(0.13)}$ | $39.09_{(0.13)}$ | $34.87_{(0.12)}$ | $40.57_{(0.15)}$ | $33.62_{(0.04)}$ | $39.11_{(0.07)}$ |
| | 32 | $64.47_{(0.19)}$ | $72.78_{(0.13)}$ | $63.91_{(0.23)}$ | $72.25_{(0.21)}$ | $67.10_{(0.30)}$ | $75.80_{(0.26)}$ | $63.91_{(0.09)}$ | $72.26_{(0.07)}$ |
| | 64 | $125.20_{(0.41)}$ | $137.15_{(0.37)}$ | $125.50_{(2.56)}$ | $136.95_{(0.69)}$ | $133.50_{(1.28)}$ | $146.55_{(1.43)}$ | $124.40_{(0.50)}$ | $136.40_{(0.50)}$ |
| | 128 | $246.95_{(0.39)}$ | $263.25_{(0.44)}$ | $246.95_{(0.76)}$ | $264.10_{(0.91)}$ | $254.80_{(6.06)}$ | $271.25_{(5.75)}$ | $245.25_{(0.44)}$ | $262.10_{(0.31)}$ |

Table 4 shows the elapsed time for the simulation. The elapsed time is proportional to the number of knots linearly, and it can be observed that approximately 0.6 seconds are required per knot. When $L = 16$, it takes around 10 seconds. Also, the elapsed time scales linearly with the number of knots and the block count, $s$. The random selection method involves more computations than the uniform method because it randomly selects data to set as knots, necessitating an additional sorting step. Due to the computational complexity of sorting knots, it results in increased elapsed time compared to the uniform method. The extra time is around 5-8 seconds when $L = 16$ and approximately 16-18 seconds when $L = 128$. In the case of (normal, normal), the uniform method is superior in accuracy and speed, but in the other three cases, the random method shows superior accuracy.

## 4.2 REAL DATA

The real data were obtained from the SNUH dataset (`https://physionet.org/content/inspire/1.2`), which includes measurements of various pre-surgery blood-related indicators such as BMI, along with eight continuous variables like pre-operative red blood cell and white blood cell counts. The dataset comprises 38,527 rows. To facilitate experimental setup, we sampled 32,768 out of the 38,527, which corresponds to one block, for correlation calculations. The vari-

able information includes BMI, preop_hb, preop_wbc, preop_plt, preop_pt, preop_aptt, preop_got, preopt_bun, and preop_cr, as listed in Table 5.

Table 5: Variable information related to blood test including BMI value.

| NAME | DESCRIPTION |
| --- | --- |
| PREOP_HB | HEMOGLOBIN LEVEL |
| PREOP_WBC | LEUKOCYTE LEVEL |
| PREOP_PLT | PLATELETS LEVEL |
| PREOP_PT | PROLONGED PROTHROMBIN TIME |
| PREOP_APTT | ACTIVATED THROMBOPLASTIN TIME |
| PREOP_GOT | LIVER FUNCTION TEST |
| PREOP_BUN | KIDNEY FUNCTION TEST1 |
| PREOP_CR | KIDNEY FUNCTION TEST2 |

We compare the accuracy and computation time of approaches involving uniform selection and randomly sampling $L$ data points. Table 6 presents the results regarding the absolute differences in Spearman rank correlation when applied to real data. A total of 36 $(= {}_9C_2)$ rank correlation coefficients were calculated for nine variables, and MAD was averaged for these 36 correlations, displaying their averages and standard deviations. As $L$ increases, the MAD values decrease. Similar to the simulation results, the random selection method with $L = 16$ knots outperforms the uniform selection method with $L = 128$. Notably, a significant difference in MAD values compared to the simulation results can be observed. This difference is attributed to the unknown distribution of the data influencing the outcomes. Given that many real datasets exhibit non-normal distributions, uniformly setting knots is less efficient for asymmetric distributions, as evident in the simulation results. In terms of elapsed time, the random selection method requires more time than the uniform selection method, primarily due to the necessity of sorting knots in the random selection method.

Table 6: Elapsed time and MAD for real data

| | MAD | | ELAPSED TIME | |
| --- | --- | --- | --- | --- |
| # OF KNOTS($L$) | UNIFORM | RANDOM | UNIFORM | RANDOM |
| 16 | $0.0835_{(0.0838)}$ | $0.0052_{(0.0054)}$ | $9.62_{(0.06)}$ | $16.33_{(0.10)}$ |
| 32 | $0.0535_{(0.0433)}$ | $0.0021_{(0.0018)}$ | $17.38_{(0.10)}$ | $27.16_{(0.16)}$ |
| 64 | $0.0433_{(0.0452)}$ | $0.0008_{(0.0007)}$ | $32.85_{(0.23)}$ | $46.66_{(0.20)}$ |
| 128 | $0.0187_{(0.0222)}$ | $0.0005_{(0.0004)}$ | $63.82_{(0.32)}$ | $82.85_{(0.29)}$ |

## 5 DISCUSSION

In this study, we introduce a method for estimating group ranks of homomorphic encryption data by estimating the ECDF comparison with pre-defined uniformly or randomly selected knots. The proposed approach effectively estimates the Spearman rank correlation, which measures the correlation between the ranks of two variables. Our simulations assessed accuracy and computational complexity across various distribution types, the number of knots, and selection methods. Additionally, we successfully demonstrated the estimation of Spearman rank correlations among the measured variables when applying this method to preoperative blood marker data.

We enumerate topics for future work. First, statistical survival analysis frequently employs ranking-based methods. These methods can be utilized in survival regression models, such as the Kaplan-Meier estimator and Cox regression, which involve estimating the cumulative survival probability up to a specific time. The group rank estimation method proposed for homomorphic encrypted data in this study can also be extended to Kendall's $\tau$ statistic, a distribution-agnostic measure of the association between two variables. Moreover, when two populations, denoted as $X_1, X_2, \ldots, X_m \sim F$, and $Y_1, Y_2, \ldots, Y_n \sim G$, are available, it is expected that the Mann-Whitney U-statistics can be used to test the null hypothesis $H_0 : \Delta = 0$, concerning the difference ($\Delta$) in means between the two populations. This is further elaborated in the appendix. We will explore these topics in our future research.

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

# A APPENDIX

## A.1 PROOF OF PROPOSITION 3.2

*Proof.* Take an element $x$ in $\mathbb{R}$. Then by the definition,

$$
\begin{aligned}
F(x;\xi) &= \sum_{j=1}^{L} I(\xi_j \le x) \frac{1}{n} \sum_{i=1}^{n} I(\xi_{j-1} \le X_i < \xi_j) \\
&= \frac{1}{n} \sum_{i=1}^{n} \sum_{j=1}^{L} I(\xi_j \le x)\, I(\xi_{j-1} \le X_i < \xi_j).
\end{aligned}
$$

Since $I(\xi_j \le x) = 1$ for $\xi_j \le x$ otherwise 0, we have

$$
F(x;\xi) = \frac{1}{n} \sum_{i=1}^{n} \sum_{\xi_j \le x} I(\xi_{j-1} \le X_i < \xi_j).
$$

Using the disjoint union property,

$$
\sum_{\xi_j \le x} I(\xi_{j-1} \le X_i < \xi_j) = I(\xi_0 \le X_i < \xi_*(x)).
$$

Since $\xi_0 = -\infty$, the right hand side is $I(\xi_0 \le X_i < \xi_*(x)) = I(X_i < \xi_*(x))$ and

$$
\begin{aligned}
F(x;\xi) &= \frac{1}{n} \sum_{i=1}^{n} I(X_i < \xi_*(x)) \\
&= \lim_{z \to \xi_*(x)-} \frac{1}{n} \sum_{i=1}^{n} I(X_i \le z) \\
&= F_n(\xi_*(x)-).
\end{aligned}
$$

Using the equality, one can easily show the inequality in Eq. equation 4. Furthermore, $F(x;\xi)$ is a consistent estimator because

$$
||F(x;\xi) - F(x)||_\infty \le ||F(x;\xi) - F_n(x)||_\infty + ||F_n(x) - F(x)||_\infty,
$$

where $\|\cdot\|_\infty$ is the $l_\infty$ norm. On the right-hand side, the ECDF $F_n$ of the second term is expected to be consistent by the strong law of large numbers (Geer, 2000). Additionally, the knot set $\xi$ of the first term becomes the original observed set as $n \to \infty$. $\square$

## A.2 PROOF OF PROPOSITION 3.3

*Proof.* By the inequality in Eq. equation 4, one can show that the first inequality $r(X_i;\xi) \le r(X_i;\eta)$ holds. Let $\eta_* = \max\{\eta_j \mid \eta_j \le X_i\}$. Then, we have

$$
F(X_i;\eta) = F_n(\eta_*-) \le F_n(X_i-) < F_n(X_i).
$$

Since $nF(X_i;\eta)$ and $nF(X_i)$ are integers satisfying $nF(X_i;\eta) < nF(X_i)$, we have $1 + nF(X_i;\eta) \le nF(X_i)$. This implies $r(X_i;\eta) \le r(X_i)$. Hence, we have $\bar{r}_\xi \le \bar{r}_\eta$ by $r(X_i;\xi) \le r(X_i;\eta)$. Suppose that $X$ has no ties. Then, the mean of $r(X_i)$ is $(n+1)/2$ and we have $\bar{r}_\eta \le (n+1)/2$. $\square$

MAN-WHITNEY U-STATISTICS

Let $F$ and $G$ be cumulative distributions. Suppose we have two populations, $X_1, X_2, \ldots, X_m \sim F$, and $Y_1, Y_2, \ldots, Y_n \sim G$, and denote the difference in the means of these two populations as $\Delta$. This discussion focuses on the method of utilizing group rank to compute Mann-Whitney U-statistics:

$$U = \sum_{i=1}^{m} \sum_{j=1}^{n} I(Y_j > X_i).$$

This is for testing the null hypothesis $H_0 : \Delta = 0$ regarding the difference in the location of the data. After combining $X_i$ and $Y_j$ to form a one-sample with size $m + n$, we can consider $\{\xi_k\}_{k=1}^{K}$ knots as follows:

$$\frac{1}{K} \sum_{k=1}^{K} \sum_{i=1}^{m} \sum_{j=1}^{n} I(Y_j > \xi_k > X_i) = \frac{1}{K} \sum_{k=1}^{K} \sum_{i=1}^{m} \sum_{j=1}^{n} I(Y_j > \xi_k) I(\xi_k > X_i)$$

$$= \frac{1}{K} \sum_{k=1}^{K} u_k v_k,$$

where $u_k = \sum_{j=1}^{n} I(Y_j > \xi_k), v_k = \sum_{i=1}^{m} I(\xi_k > X_i)$.

## A.3 KENDALL'S TAU

When there is a positive correlation between two variables in bivariate data $X_i, Y_{i i=1}^{n}$, choosing two observations, $(X_i, Y_i)$ and $(X_j, Y_j)$, typically results in one point appearing in the lower left, while the other tends to be in the upper right. In such cases, an increase in one variable corresponds to a rise in the other, indicating a concordant relationship. In contrast, it can be referred to as a discordant relationship in the opposite situation.

For any two independent pairs of observations $(X_i, Y_i), (X_j, Y_j)$, Kendalls' tau statistic is defined by

$$\tau = p_c - p_d,$$

where $p_c = \Pr((X_i - X_j)(Y_i - Y_j) > 0)$ and $p_d = \Pr((X_i - X_j)(Y_i - Y_j) < 0)$. Define $A_{ij} = \text{sign}(X_j - X_i)\text{sign}(Y_j - Y_i)$. The marginal probability distribution of the $A_{ij}$ is

$$f_{A_{ij}}(a_{ij}) = \begin{cases} p_c & \text{if } a_{ij} = 1 \\ p_d & \text{if } a_{ij} = -1 \\ 1 - p_c - p_d & \text{if } a_{ij} = 0 \end{cases}.$$

Then $\mathrm{E}(A_{ij}) = p_c - p_d$. There are only ${}_nC_2$ sets of pairs that needs to be considered. An unbiased estimator of $\tau$ is provided by

$$T = \sum_{i<j} \frac{A_{ij}}{{}_nC_2} = 2 \sum_{i<j} \frac{A_{ij}}{n(n-1)}.$$

Let $u_{ij} = \text{sign}(X_i - X_j), v_{ij} = \text{sign}(Y_i - Y_j)$, and $a_{ij} = u_{ij}v_{ij}$ for all $i, j$. Assuming that $x_i \neq x_j$ and $y_i \neq y_j$ for all $i \neq q$, we have

$$\sum_{i=1}^{n} \sum_{i=1}^{n} u_{ij}^2 = \sum_{i=1}^{n} \sum_{i=1}^{n} v_{ij}^2 = n(n-1).$$

If we use pre-specified knots $\{\xi_j^X\}_{j=1}^{K}$ and $\{\xi_j^Y\}_{j=1}^{K}$ instead of $\{X_j\}_{j=1}^{n}$s and $\{Y_j\}_{j=1}^{n}$s, respectively,

$$p_c = \Pr((X_i - \xi_j^X)(Y_i - \xi_j^Y) > 0) \text{ and } p_d = \Pr((X_i - \xi_j^X)(Y_i - \xi_j^Y) < 0).$$

