# OpenReview forum: "Group rank for encrypted data"
_ICLR.cc/2025/Conference — Submitted to ICLR 2025_

### Official Review · Reviewer_mFvL · 2024-10-30

**Soundness:** 3
**Presentation:** 2
**Contribution:** 2
**Rating:** 5
**Confidence:** 3

**Summary:**

This paper proposes a new method for approximate computation based on the group rank of homomorphic encryption. The authors introduce a grouped rank within HE, and propose new method to estimating Spearman rank correlation in two-dimensional space with HE. Finally, the authors conduct empirical studies to validate the effectiveness of the proposed method.

**Strengths:**

The main advantages can be listed as follows:

1.	The paper addresses the new problem of HE-friendly computations based on ranks and proposes a new method for estimating Spearman rank correlation using HE with a group rank approach, supported by theoretical analysis.
2.	The paper conducts experiments on simulated and real datasets to show the effectiveness of the proposed method.

**Weaknesses:**

Despite the strengths, there are some issues with the paper as follows:

1.	I am concerned about the paper's contributions. The proposed method for computing Spearman rank correlation on two-dimensional data using HE operations may have limited impact due to the dimensional constraints. This limitation could reduce the contribution and practical relevance of the work.
2.	I am also concerned about the validity of the experimental evaluation. The authors only compared the performance and latency of the proposed method under various hyper-parameters. Including comparisons with other baseline methods across additional datasets would significantly enhance the evaluation of the proposed method's effectiveness and robustness.

**Questions:**

1.	Are there potential applications of the proposed method in machine learning tasks?
2.	In Line 63, the authors mention that the precise ranking method in HE requires $O(n^2)$ complexity. What is the computational complexity of the proposed method?
3.	Can the proposed method be extended to data with dimensions greater than two?

---

### Official Review · Reviewer_NvpQ · 2024-11-02

**Soundness:** 3
**Presentation:** 3
**Contribution:** 2
**Rating:** 5
**Confidence:** 3

**Summary:**

This paper proposes a group ranking method based on homomorphic encryption, aiming to reduce the computational complexity of encrypted data sorting. By using approximate ranking to reduce the number of comparisons, it shows the application in calculating Spearman rank correlation for bivariate variables.

**Strengths:**

1. A group ranking method based on homomorphic encryption is proposed to reduce the computational complexity of sorting encrypted data.
2. Through a series of numerical experiments, the method has been verified to be effective under different data distributions.
3. The proposed method provides a new approach for privacy-preserving machine learning.

**Weaknesses:**

1. Lack of Comparisons with Related Work. The paper refers to some existing homomorphic encryption methods but lacks in-depth comparisons, particularly with existing solutions for encrypted data sorting. Furthermore, the most recent work cited in the related work section is from seven years ago, missing recent advancements that could help highlight the improvements made by the proposed method.
2. Insufficient Detail in Experimental Section. The experimental section lacks detailed discussions on error sources and dataset specifics, such as the differences in results under different data distributions, and why the "random selection" method outperforms the "uniform selection" method in some cases. Additionally, there is insufficient description of the statistical significance of experimental results (e.g., standard error or confidence intervals), which is important for assessing the reliability and stability of the results.
3. Lack of Diversity in Real-World Data Experiments. The real-world data experiment only used a single dataset to estimate the Spearman rank correlation, lacking diverse application scenarios. Therefore, the generalizability of the method and its performance in other real-world contexts have not been fully demonstrated.

**Questions:**

1. What is the advantage of the proposed method comparing to existing methods?
2. What is the reasonability of the related descriptions in experimental section?

---

### Official Review · Reviewer_eLh9 · 2024-11-04

**Soundness:** 2
**Presentation:** 2
**Contribution:** 2
**Rating:** 3
**Confidence:** 5

**Summary:**

The paper considers the problem of computing rank correlation between two collections of data samples over homomorphically encrypted data. It seeks to reduce the $\mathcal{O}(n^2)$ computational complexity of computing such rank correlations. The paper proposes an interesting solution for this purpose. However, the paper's topic may not be suitable for the conference due to limited machine-learning contributions. Even from a security/privacy viewpoint, the lack of a threat model and security analysis limits the paper's contributions.

**Strengths:**

- Computing rank correlations between sets of data points is a problem of general interest in science and engineering. Sometimes, one has to compute this over sensitive data. Homomorphic encryption is an attractive solution for preventing unauthorized access to the underlying data.
- The proposed solution reduces the complexity from $\mathcal{O}(n^2)$ to $\mathcal{O}(Ln)$ for some fixed number of knots $L$.
- Results on synthetic and one real dataset show that the proposed approach can accurately estimate the Spearman Rank Correlation.

**Weaknesses:**

- This paper is not suitable for the ICLR audience. It is focused on adapting Spearman rank correlation for evaluation over Homomorphically Encrypted data. In this reviewer's opinion, the paper has minimal to no machine learning innovation.
- There is no security analysis. The security protocol, parties participating, and the threat model have not been discussed. These are critical for any paper focused on privacy and security.
- The evaluation is limited to a single real dataset. So, it is unclear if the performance benefits are generalizable to other applications.

**Questions:**

- Figure 1 illustrates an example of face recognition using Spearman rank correlation. However, there is no evaluation of the proposed rank correlation for face recognition. Showing this application and comparing it to prior work [a-b] would strengthen the paper and enhance its relevance to ICLR.

[a] HERS: Homomorphically Encrypted Representation Search, TBIOM 2022
[b] Fast and Accurate Biometric Search Under Encryption, 2023

**Details Of Ethics Concerns:**

There are no ethical concerns for this paper.

---

### Official Review · Reviewer_RDkK · 2024-11-04

**Soundness:** 2
**Presentation:** 1
**Contribution:** 2
**Rating:** 3
**Confidence:** 4

**Summary:**

In data analysis, using homomorphic encryption schemes to compute rankings requires comparison operations, which may introduce significant computational costs. Currently, there is no efficient solution to this challenge. To address this issue, the authors propose an approximate ranking method that leverages pairwise comparisons between data points to derive the rankings of encrypted information. The authors conducted numerical experiments to validate their method, demonstrating that the difference in values between rank correlation and approximate rank correlation is negligible.

**Strengths:**

The authors of this paper devised a practical application for a fully homomorphic encryption (FHE) scheme in the realm of data analysis. They introduced an approximate rank method to mitigate computational overhead. Furthermore, they leveraged the batching capabilities of the CKKS scheme to enable efficient batch processing of encrypted data.

**Weaknesses:**

The authors should provide a more detailed and clear explanation of how the CKKS scheme is employed to complete the data analysis task. For example, since polynomial approximations are necessary for comparison operations, the authors should elaborate on the process of selecting the degree of these polynomials and the impact of this choice on the accuracy of the data analysis.

The experimental comparisons and analyses need to be reorganized, as the current presentation is very unclear. Furthermore, there are some typos, such as the incorrect notation CKSS. And the parameter settings for the CKKS scheme should also be described more accurately and comprehensively. Ring dimension should be replaced by poynomial degree, and formula of "The number of slots is 32768 = log(ring dimension/2)" is incorrect.

**Questions:**

What is the security model in this paper? If it is a server-client model, is there still a need to use ciphertext-to-ciphertext comparison operations? The author should provide a clearer description of the background.

---

### Official Review · Reviewer_5vdF · 2024-11-08

**Soundness:** 2
**Presentation:** 3
**Contribution:** 2
**Rating:** 5
**Confidence:** 4

**Summary:**

This paper proposes a method to improve computational efficiency in rank analysis of encrypted data. Traditional methods for calculating ranks on homomorphically encrypted (HE) data are significantly time-consuming, especially for computing Spearman rank correlation. This study introduces a novel "group rank" approach, which approximates ranks by focusing on comparisons at specifically selected points, thereby reducing the number of comparisons and computational cost.
The effectiveness of the proposed method was validated through simulations and real data analysis, examining various data distributions and numbers of comparison points. Experiments using the SNUH dataset successfully estimated correlations among variables in actual test data, demonstrating the practical applicability of this approach.

**Strengths:**

The strengths of this paper lie in its innovative approach to bypassing the complexity of rank calculation on encrypted data by introducing an efficient computation method based on group rank rather than traditional sorting techniques. By incorporating the concept of group rank, the authors present a robust method for estimating ranks in large-scale homomorphically encrypted datasets.
This study addresses the challenge of estimating Spearman rank correlation in a two-dimensional space and proposes a method to minimize estimation error. The paper introduces a novel approach to statistical data processing within homomorphic encryption, contributing valuable insights to the fully homomorphic application field.

**Weaknesses:**

While this paper introduces a homomorphic encryption-based method for processing statistical data, its focus is somewhat outside the primary scope of the ICLR, aligning more closely with applications of homomorphic encryption per se. Additionally, assessing the suitability of the proposed approach as a general method for the statistical processing of encrypted data will be required. Although the rank-based comparison method offers some gains in comparison, it remains uncertain whether this approach can be generalized effectively for other types of statistical processing.

**Questions:**

Can the proposed method be applied to other homomorphic statistical processing techniques? Is the ECDF computed in plaintext or encrypted? If it is calculated in plaintext, does this process present any privacy concerns?

**Details Of Ethics Concerns:**

There is no concern.

---

### Meta-Review · Area_Chair_C5rj · 2024-12-06

**Metareview:**

The paper considers the problem of computing rank correlations for homomorphically encrypted data. The main contribution is an innovative "group rank" approach, which approximates ranks by focusing on comparisons at specifically selected points. The reviewers praised the novelty of the approach and the reduction in complexity achieved by it. However several serious issues were raised, including fit to ICLR, little clarity in the presentation, absence of a security analysis and limited evaluation. The authors have not provided any response, and these concerns still stand. Thus, I think the paper should be rejected in its current form.

**Additional Comments On Reviewer Discussion:**

Unfortunately, the authors did not respond to the comments of the reviewers, so there was no discussion.

---

### Decision · Program_Chairs · 2025-01-22

Reject